# Is there proactive inhibitory control during bilingual and bidialectal language production?

**Mathieu Declerck[1], Elisabeth Özbakar[1], Neil W. Kirk[2]***

**1** Vrije Universiteit Brussel, Brussels, Belgium, **2** Abertay University, Dundee, Scotland, United Kingdom

* n.kirk@abertay.ac.uk

**Data Availability Statement:** All data and analysis files are available from the Open Science Framework (https://osf.io/rdwgp/).

**Funding:** This research is funded by a Carnegie Trust for the Universities of Scotland Research

## Abstract

The bilingual language control literature generally assumes that cross-language interference resolution relies on inhibition of the non-target language. A similar approach has been taken in the bidialectal language control literature. However, there is little evidence along these lines for proactive language control, which entails a control process that is implemented as an anticipation of any cross-language interference. To further investigate the possibility of proactive inhibitory control, we examined the effect of language variety preparation time, by manipulating the cue-to-stimulus interval, on parallel language activation, by manipulating cognate status. If proactive language control relies on inhibition, one would expect less parallel language activation (i.e., a smaller cognate facilitation effect) with increased proactive inhibitory control (i.e., a long cue-to-stimulus interval). This was not the case with either bilinguals or bidialectals. So, the current study does not provide evidence for proactive inhibitory control during bilingual and bidialectal language production.

## Introduction

A prominent assumption regarding bilingual language control, which entails a process that reduces cross-language interference during bilingual language processing and increases the chances of selecting words in the target language, is that it relies on inhibition of the non-target language (e.g., [1–3]). A similar assumption has been put forward regarding language processing by bidialectals (i.e., speakers of a regional dialect that are also fluent in a standard language variety). However, most of the evidence for inhibitory control with bilinguals (e.g., [4–6]), and especially with bidialectals [7–9], relates to reactive language control, which entails a control process that is implemented when cross-language interference is detected. As can be seen in a recent review on proactive language control [10], which is a control process implemented as an anticipation of any cross-language interference, most measures of proactive language control are explained with inhibition, but very little direct evidence has been put forward in favor of proactive language control mainly relying on inhibition. In the current study, we set out to further investigate the possibility of proactive language control mainly relying on inhibition during both bilingual and bidialectal language production.

Proactive language control has been investigated through several effects in the bilingual field (for a review, see Declerck [10]). For instance, the reversed language dominance effect

Incentive Grant (https://www.carnegie-trust.org/) awarded to NWK (RIG009864). The funders had and will not have a role in study design, data collection and analysis, decision to publish, or preparation of the manuscript.

**Competing interests:** The authors have declared that no competing interests exist.

(e.g., [11–14]), which entails worse second language (L2) performance than first language (L1) performance in mixed language blocks. This effect has typically been explained based on inhibition: To make the activation levels of both languages similar, which should improve overall performance in mixed language blocks, L1 is proactively inhibited (e.g., [15]). However, a similar account of the effect could also rely on proactive increased activation of L2 [16].

Proactive language control has also been investigated in single language blocks with the blocked language order effect, which is characterized by worse behavioral performance in a single language block when previously producing in a single language block that required using a different language (e.g., [17–20]). This effect is typically obtained in one of two ways: One setup relies on two consecutive single language blocks in which the bilingual participants usually name pictures in language A in Block 1 and in language B in Block 2. Behavioral results typically show worse performance in Block 2 (e.g., [17,21]. The other setup relies on three consecutive single language blocks, with Blocks 1 and 3 requiring naming in language A and Block 2 requiring naming in language B. The behavioral pattern in this setup entails worse performance in Block 3 than in Block 1 (e.g., [17,18]).

The blocked language order effects have typically been explained with inhibition. More specifically, when using Language A in a single language block, Language B will be proactively inhibited throughout. When the next single language block requires production in Language B, performance will be worse as the proactive inhibition on Language B from the previous block is assumed to persist [17]. However, a recent study also proposed an activation account of this effect [20]: It could very well be that consistently using Language A increases the activation of Language A to the point that items of this language will still be highly activated when a different language (Language B) is used in the next block. These considerable competitors from Language A, while Language B is the target language, would reduce performance in single language blocks presented after a single language block in a different language.

ERP blocked language order studies directly examined the possibility of proactive inhibitory control through the N2, which is a negative-going peak usually found around 200–350 ms after stimulus presentation that has been linked to inhibition (e.g., [22]). Misra and colleagues [19], for instance, examined the blocked language order effect with Chinese-English bilinguals. Half of the participants were asked to name pictures in Chinese in the first two blocks, followed by two blocks that required producing in English, and vice versa for the other half of participants. Their results showed a larger N2 in the third L1 block than in the first, indicating increased inhibition in Block 3. However, in the L2 blocks, a larger N2 was observed in the first than in the third block. Moreover, while the blocked language order effect interacted with laterality, follow-up analyses on each of the hemispheres showed no significant N2 blocked language order effect. Other ERP studies that investigated the blocked language order effect also did not show straightforward evidence for proactive inhibitory control. Branzi et al. [17], for instance, found no blocked language order effect with the N2. While Wodniecka et al. [20] did find a larger negativity in the N2 time window for the second single language block after previously producing in another language, this was interpreted in terms of the N300 (i.e., a negative-going peak typically found around 250–400 ms that has been linked to difficult to interpret pictures and relatedness to the previous stimulus; e.g., [23,24]), not the N2. So, these ERP studies do not provide unequivocal evidence in line with the notion of proactive language control relying on inhibition.

Recently, Branzi and colleagues [25] set out to investigate the issue of proactive inhibitory control in mixed language blocks by examining the influence that language preparation time has on cross-language activation. In this study, Spanish-Basque-English multilinguals had to name pictures in mixed language blocks that required a language switch on every trial between their first language (L1; Spanish) and their third language (L3; English). Additionally, a cue,

which indicated the language to be used, was presented prior to each picture. The interval between the cue and picture could either be long (1000 ms), allowing for a lot of language preparation time and thus a lot of proactive language control, or short (150 ms), allowing for less language preparation time and thus less proactive language control. An additional manipulation was that half the pictures were cognate words between L1 and L3 and the other half were non-cognates. Cognates are words with a similar form/phonology across two languages and are typically processed faster than noncognates because the representations of both languages are activated leading to facilitation at the phonological level (e.g., [11,25]). The logic behind the cue-to-stimulus interval (CSI) manipulation and the cognate status manipulation was that if proactive inhibitory control was implemented during the interval between the cue and the picture, it would reduce the activation of the non-target language. In the case of cognates, this would reduce performance, since the non-target language does not substantially boost the target phonemes anymore. So, if proactive language control, instigated during the CSI, relies on inhibition, one would expect a decrease of the cognate facilitation effect. This decrease would be more pronounced with a long interval between the cue and picture, since more proactive inhibitory control could be implemented in this condition. Whereas the fMRI results were along these lines, with the bilateral anterior ventrolateral prefrontal cortex showing a reduction in the long CSI relative to trials with a short CSI for the cognate facilitation effect, the behavioral results were not. More specifically, no interaction was observed between CSI and cognate status in the error rates. It could of course be that the dependent variable in the behavioral analysis (i.e., error rates) did not allow to find such an effect, as language control measures are typically more robust in reaction time (RT) analyses than in error analyses.

In the current study, we set out to further examine the possibility of proactive inhibitory control by relying on the logic put forward in Branzi et al. [25], with a smaller cognate facilitation effect with a long than short CSI providing evidence for proactive inhibitory control. Unlike Branzi and colleagues [25], however, we would mainly rely on RT analyses. Furthermore, we would also investigate bidialectals (Experiment 2), next to bilinguals (Experiment 1), since even less is known about proactive inhibitory control during bidialectal language production. Several studies have provided evidence for reactive inhibitory control during bidialectal language production, through asymmetrical switch costs (i.e., larger costs for switching to the more dominant variant than to the less dominant variant; [7–9]), but very little research has gone into proactive inhibitory control with bidialectals.

## Experiment 1

### Method

Next to the hypothesis, the sample size and analyses of both experiments were all pre-registered (https://osf.io/rdwgp/). Furthermore, the data can also be found with this Open Science Framework link. The materials for both experiments can be accessed as Gorilla Open Materials (Exp1: https://app.gorilla.sc/openmaterials/236318; Exp2: https://app.gorilla.sc/openmaterials/241271). Both experiments received approval by Abertay University's ethics committee (EMS3231).

### Participants

Forty-one Dutch-English bilinguals were recruited. One participant was excluded due to recording failure and another three participants were excluded at the accuracy coding stage due to the presence of extraneous background noise, leaving 37 participants (23 participants identified as women and 14 as men) with an average age of 27.3 years (SD = 8.8). Following the experiment, the participants were asked to fill in a language background questionnaire

**Table 1. Overview of the demographic information (SD in brackets) for participants in the Dutch-English (Experiment 1) and Dundonian-English (Experiment 2) experiments.**

|  | Experiment 1 Dutch-English | Experiment 2 Dundonian-English |
|---|---|---|
| English current use (%) | 26.0 (24.1) | 57.9 (31.5) |
| Dutch/Dundonian current use (%) | 65.0 (31.1) | 42.1 (31.5) |
| English LexTale (%) | 72.8 (15.4) | 90.8 (9.5) |
| Dutch Lextale (%) | 82.3 (12.7) | N/A |

(cf. [7]), and English and Dutch vocabulary tests based on lexical decision tasks (i.e [26]; see Table 1).

## Stimuli

Seventy-two concrete items that could be named in Dutch and English were used. Pictures for these items were sourced from the MultiPic picture set [27] and were depicted as 300x300 pixel grayscale images. Half of the items depicted cognates items between the two languages (e.g., "apple", which is "appel" in Dutch) and half depicted non-cognate items (e.g., "horse", which is "paard" in Dutch; see Appendix for the full list). The cognate (Dutch: 1.4; English: 1.4) and non-cognate (Dutch: 1.4; English: 1.4) items had on average a similar number of syllables and a similar Dutch ([28]; cognate: 78.81; non-cognate: 72.50) and English ([29]; cognate: 79.83; non-cognate: 82.79) word frequency per million (all $p$'s >.77).

To indicate which language participants had to use on each trial, colored frames (i.e., green and blue) were used. The color-to-language assignment was counterbalanced across participants.

## Procedure

A mixed-language picture naming study was presented online using the Gorilla Experiment Builder platform (http://gorilla.sc; for a review, see [30]). This experiment was limited to participants using PC desktop or laptop devices. After providing their informed consent and completing a microphone check, participants were given general instructions for the task, including their color-to-language assignment for the cues, and were presented with a 7-trial demonstration of a Dutch-English bilingual naming pictures in both languages. Participants then completed a 10-trial practice block where they were cued to name English and Dutch items in an unpredictable sequence, with half of these items containing a long CSI of 1250ms (cf. [31,32]), and half presenting the cue and stimulus simultaneously (i.e., no CSI). The long CSI used in the current study was longer than most studies that manipulated CSI length (e.g., [6,12,25,33–36]), with the exception of Lavric et al. [37] that used a long CSI of 1500ms. We chose to use no CSI, instead of a very short CSI (e.g., 100ms CSI), along the lines of previous studies [35,38], as this would provide a more extreme difference between CSI conditions (i.e., preparation vs. no preparation relative to more vs. less preparation).

On completion of the practice block, participants moved to the main experiment, which consisted of two blocks with varying CSI lengths (0 and 1250ms). In the no CSI block (0 ms), participants were instructed that the colored border (indicating the language they should name the item in) and the stimulus would appear at the same time, and in the long CSI block (1250ms) they were instructed that the colored border cue would also appear prior to the picture. The order of CSI block types was counterbalanced across participants.

Each block contained 72 trials. A pseudo-randomized sequence list was used to ensure an equal number cognate and non-cognate items were presented across language-switch trials

(i.e., trials that use a different language than the prior trial) and language-repetition trials (i.e., trials that use the same language as the previous trial) for both language varieties. Each participant received a version of the list that started with two English trials in one block, and two Dutch trials in the other block, which was counterbalanced across CSI blocks and participants. Moreover, each of the pictures occurred twice across the experiment, once in each CSI block and always in a different language.

Each trial in the long CSI block started with a fixation cross that appeared in the center of the screen for 250ms. This was followed by a colored language cue for 1250ms, after which the target picture was introduced in the cue. The picture and cue were presented together for 2500 ms, during which the participants' vocal responses were captured.

Trials in the no CSI block were similar, except that there was no presentation of the language cue prior to the picture. Furthermore, in order to keep the overall trial length consistent between blocks, a final blank screen of 1250 ms was presented after the presentation of cue and picture in the no CSI block.

## Analysis

**Accuracy coding.** Participants vocal responses were captured as individual audio files on a trial-by-trial basis and downloaded at the end of the experiment. These sound files were then uploaded into a new accuracy coding task on the Gorilla Experiment Builder platform, alongside a new task spreadsheet containing information about each trial (e.g., target language, target word, trial type, cognate status, and corresponding sound file name). This new accuracy coding task displayed the target language, the target word, the participants' anonymized ID number, and offered three response buttons: Correct (for items produced correctly in the target language), Incorrect (for items either produced with the wrong name or the correct name but in the wrong language), and No/Other Sound (for items where no sound was detected, or extraneous background noise would interfere with response time detection). A version of this Accuracy Coding Task is available as Open Materials on the Gorilla Platform: (https://app.gorilla.sc/openmaterials/236318).

**Reaction time extraction.** Participants' audio response files were converted from.weba to.wav using VLC media player [39] and were then uploaded to the Chronset system [40] for RT detection. This was then combined with the accuracy data for analysis. To ensure the validity of the Chronset system, we manually measured a random selection of 5% of each participants' trials (across both experiments). A Pearson correlation test showed a strong positive correlation between Chronset and manually measured RTs, $r(419) = .921$, $p < .001$.

**Outliers.** Error trials and trials where no or another extraneous sound were detected were excluded from RT analyses, as were "recovery" trials following an error trial. Furthermore, RTs under 150ms, or with RTs three standard deviations above the participant mean were discarded as outliers. Taking these criteria into account, a total of 26.5% of the RT data were excluded from analysis.

**Linear mixed-effects.** The RT data were analyzed using linear mixed-effects regression modeling [41]. Both participants and items were considered random factors with all fixed effects and their interactions varying by all random factors, using the maximal random effects structure that would result in model convergence [42].

## Results and discussion

### Error rates

Overall, error rates reached 3.6% of valid trials, thus we did not conduct any analysis in line with our pre-registered threshold of 5% errors.

**Table 2. Parameter estimates and results of significance tests in mixed-effects models for Experiment 1.**

| Dutch-English Model: RT ~ CSI * Variety * CognateStatus + (1 + CSI * Variety * CognateStatus \| Participant) + (1 + CSI * Variety \| Picture) | | | | |
|---|---|---|---|---|
| **Fixed effects** | **β** | **SE** | **t** | **P** |
| Intercept | 1137.0 | 38.1 | 29.8 | < .001 |
| CSI | 20.8 | 8.2 | 2.5 | 0.02 |
| Variety | 5.0 | 10.1 | 0.5 | 0.63 |
| Cognate Status | 32.6 | 10.2 | 3.2 | < .001 |
| CSI x Variety | -7.1 | 5.4 | -1.3 | 0.19 |
| CSI x Cognate Status | 0.8 | 6.1 | 0.1 | 0.89 |
| Variety x Cognate Status | -9.7 | 5.6 | -1.7 | 0.09 |
| 3-way interaction | -3.1 | 4.9 | -0.6 | 0.52 |

## Reaction times

RTs were analyzed using a mixed-effect linear model with CSI Length (No vs. Long CSI), Variety (Dutch vs. English), Cognate Status (cognates vs. non-cognates), and all interactions between these factors as centered fixed effects, and random effects of Participants and Items. This model converged and yielded a main effect of CSI Length, with slower responses in the No CSI condition (1131.0 ms) than in the Long CSI condition (1092.6 ms; see Table 2). This pattern is in line with many, but not all (e.g., [25]), bilingual studies that investigated the effect of CSI in mixed language blocks (e.g., [36,43]), and could be taken as evidence that increased proactive language control leads to better performance. Furthermore, along the lines of most bilingual studies investigating cognates (e.g., [11,44,45]), a significant main effect of Cognate Status was found, with faster responses with cognates (1085.9 ms) than with non-cognates (1139.7 ms). Crucially, we did not find a significant interaction between CSI Block and Cognate Status (see Fig 1), indicating that the cognate facilitation effect was not reduced by the longer preparation time in the long CSI condition.

As indicated in the pre-registration, we also wanted to make sure that Trial type (switch vs. repetition trial) did not hide any effects related to our interaction of interest (CSI Length x Cognate Status). An additional analysis with Trial type (switch vs. repetition trial) included showed worse performance in switch (1148 ms) than repetition (1076 ms) trials (i.e., switch costs), $b = 41.5$, SE = 5.9, $t = 7.1$, $p < .001$, but Trial type did not significantly interact with any other factor (all $p$'s >.11).

## Experiment 2

In experiment 2, we extended this investigation to include "bidialectal" speakers of two closely-related varieties. This would allow us to generalize the results of Experiment 1 to a group that are more balanced across their language pairs (see Table 1). Furthermore, one might assume that bilinguals and bidialectals rely on similar control processes when producing language. However, a multitude of studies have shown differences in language control across, for instance, modalities (e.g., [46]) and language pairs (e.g., [47,48]) within and across participants. So, language control is not a uniform process. Especially interesting for the current study is that a recent study even showed that very similar language pairs do not always require language control [49]. Hence, differences might be expected across bilingual and bidialectals regarding language control, as the language pairs of the latter are generally far more similar.

In this experiment, we tested participants proficient in Scottish Standard English and Dundonian Scots, a dialect spoken in and around the city of Dundee, Scotland. Scottish Standard

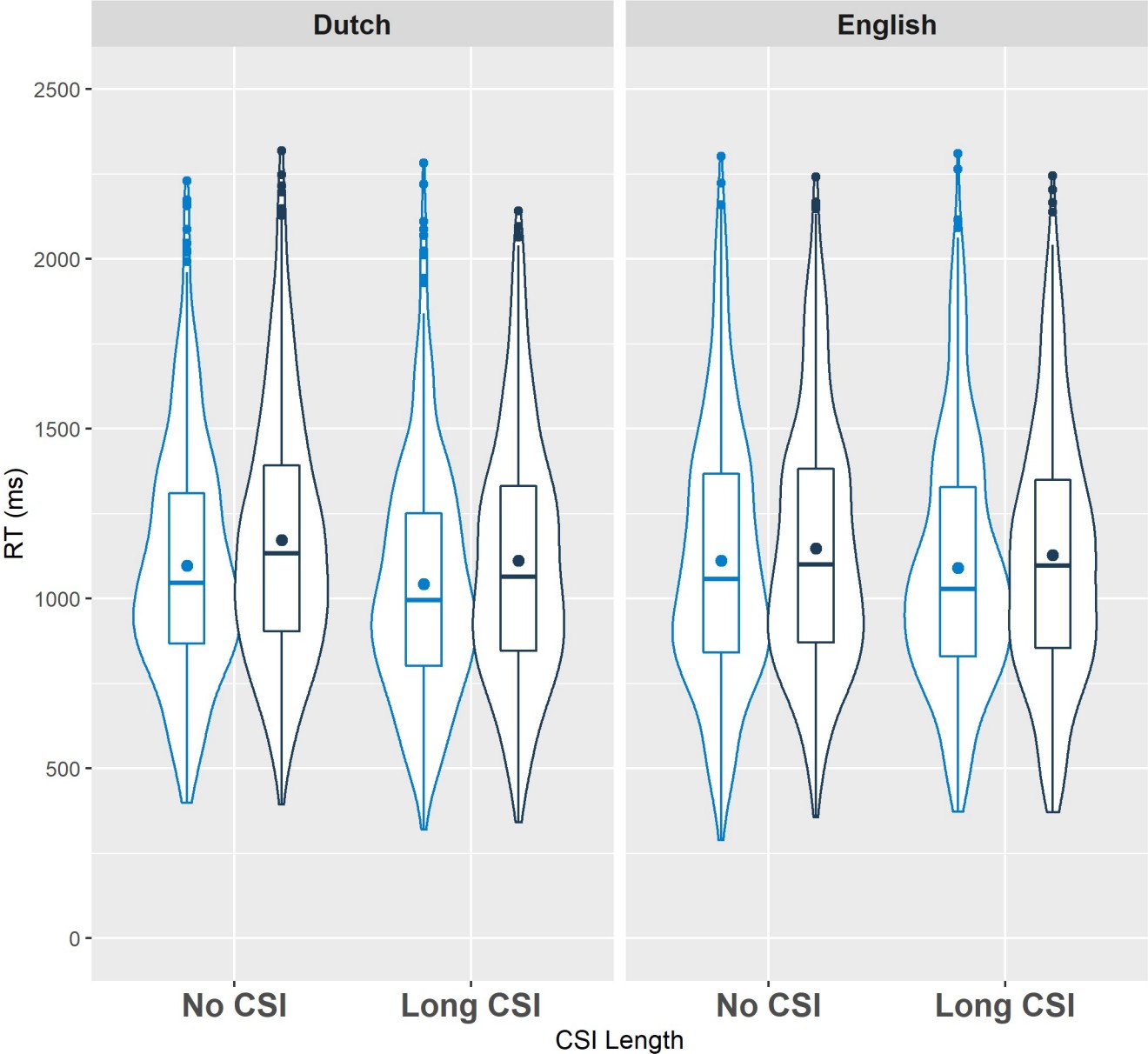

**Fig 1. Violin plot showing the distribution of cognate and non-cognate RTs in each CSI block across both varieties in Experiment 1 (Dutch/English bilinguals).** The boxplots show the interquartile range, the horizontal line represents the median, and the dot indicates the mean for each condition.

English is the standard variety of English spoken in Scotland (sometimes simply considered "English with a more or less Scottish accent"; [50]), which exists in a diglossic and linguistic continuum with Scots and its many regional varieties. Scots is a West Germanic language variety recognized by the European Charter for Regional or Minority Languages (not to be confused with Scottish Gaelic), but whose status as a separate language from English is often questioned because of their similarity [51]. Indeed, many speakers themselves consider Scots not as a language, but "just a way of speaking" [52]. The overlap between these varieties, as

well as the different status afforded to them, means that these participants are traditionally categorized and self-identify as "monolingual" [7,53]. Yet, navigating between two linguistic systems has shown to require similar processes as utilized by bilinguals (e.g., [7–9,54]), motivating their inclusion in this investigation.

So far only one study has shown any evidence for proactive language control during bidialectal language production [8]. In this study, Orcadian-English bidialectals performed a picture naming task in mixed and single language blocks. The results showed substantial mixing costs, which has been taken as a measure for proactive language control among others, such as the cognitive cost to monitor and maintain multiple languages (e.g., [55,56]; for a discussion, see [10]). So, while there is some initial evidence for proactive language control, no research has gone into the possibility that proactive language control implemented during bidialectal language production might rely on inhibition. Hence, this is the first study to investigate whether proactive language control relies on inhibition during bidialectal language production.

## Method

### Participants

Forty-six bidialectals of Scottish Standard English (henceforth English) and Dundonian Scots were recruited. Six participants were excluded at the accuracy coding stage as four had recording failures and two participants continually swapped the language variety color cues around. This left forty participants (25 identified as women, 14 as male and 1 as nonbinary) with an average age of 38.0 years (SD = 11.6). Following the experiment, the participants were asked to complete a language background questionnaire (cf. [7]), and an English vocabulary test on the basis of a lexical decision task (i.e., LexTale; [26]; see Table 1).

### Stimuli

As English and Dundonian Scots overlap considerably, fewer items were used than in Experiment 1. Eighteen pictures were used that depict concrete objects that could be named in both varieties. Half depicted cognates (e.g., "house", which is "hoose" in Dundonian Scots) while half depicted non-cognates (e.g., "children", which is "bairns" in Dundonian Scots; see Appendix for the full list). The pictures were 300x300 pixel greyscale images sourced from the Multi-Pic picture set [27] and from a previous experiment [7]. The cognate (English: 4.8; Dundonian Scots: 4.7) and non-cognate (English: 4.7; Dundonian Scots: 4.2) items had on average a similar number of syllables and a similar English ([57]; cognate: 126.0 non-cognate: 104.7) word frequency per million (all $p$'s >.27). No word frequency information was available for Dundonian Scots. Hence, we could not match the items on this characteristic.

### Procedure

The same picture naming task and procedure used in Experiment 1 was adapted for use with Dundonian-English bidialectals in Experiment 2, except for the following changes: After providing informed consent and completing the microphone check, participants were given an additional familiarization block in which all eighteen items were presented alongside the corresponding English and Dundonian Scots labels. However, as Dundonian Scots has no standardized written format, participants were given the option of hearing the Dundonian items named by a local speaker.

As in Experiment 1, each block contained 72 trials, but in Experiment 2 each item appeared four times per block. A similar pseudo-randomization as in Experiment 1 was used to ensure an equal number of cognate and non-cognate items were presented across switch and

repetition trials for both language varieties, with each item appearing once across each combination of variety (Dundonian Scots and English) and trial type (switch and repetition trials) within a block. Each participant received a version of the list that started with two English trials in one block, and two Dundonian trials in the other block, which was counterbalanced across CSI blocks and participants.

### Analysis

The analysis information was identical to that of Experiment 1. Hence, trials were subject to the same exclusion criteria as Experiment 1, which resulted in a total of 22.7% trials being excluded from analysis.

## Results and discussion

### Error rates

Error rates for this experiment reached 3.8% of valid trials, thus we did not analyze error rates in line with our pre-registered analysis plan.

### Reaction times

RTs were analyzed using a mixed-effect linear model with CSI Length (No vs. Long CSI), Variety (Dundonian vs. English), Cognate Status (cognates vs. non-cognates), and all interactions between these factors as centered fixed effects, and random effects of Participants and Items. This model converged and showed a significant main effect of Variety, with slower responses in English (1072.2 ms) than in Dundonian Scots (1006.7 ms; see Table 3). Along the lines of previous studies with bidialectals [7–9], a significant main effect of Cognate Status was found, with faster responses with cognates (978.9 ms) than with non-cognates (1100.6 ms). However, unlike Experiment 1, no significant effect of CSI Length was observed. This could be interpreted as little to no proactive language control being implemented during the CSI by bidialectals, which could be due to the large overlap between the two language variants (cf. [49]), especially compared to the language pairs of Experiment 1. Yet, while no evidence for bidialectal proactive language control with the CSI length manipulation was observed in the current study, Kirk et al. [8] did find some evidence for such a process during bidialectal language production through mixing costs. It could be that the difference in setup or even the type of bidialectals (Dundonian-English vs. Orcadian-English) resulted in this discrepancy across studies. Another possibility is that the mixing costs observed in Kirk et al. [8] were not so much due to

**Table 3. Parameter estimates and results of significance tests in mixed-effects models for Experiment 2.**

**Dundonian-English Model: RT ~ CSI \* Variety \* CognateStatus + (1 + CSI \* Variety \* CognateStatus | Participant) + (1 + CSI \* Variety | Picture)**

| Fixed effects | β | SE | t | p |
|---|---|---|---|---|
| Intercept | 1048.4 | 30.8 | 34.1 | < .001 |
| CSI | 4.6 | 8.4 | 0.6 | 0.58 |
| Variety | 33.8 | 8.7 | 3.9 | < .001 |
| Cognate Status | 62.7 | 10.7 | 5.9 | < .001 |
| CSI x Variety | 7.6 | 4.3 | 1.8 | 0.08 |
| CSI x Cognate Status | -0.9 | 4.4 | -0.2 | 0.85 |
| Variety x Cognate Status | 29.2 | 8.1 | 3.6 | < .001 |
| 3-way interaction | 4.0 | 3.5 | 1.1 | 0.26 |

proactive language control, but because of maintaining and/or monitoring both language vari-
ants in mixed language variety blocks relative to single language variety blocks.

There was also an interaction between Variety and Cognate Status, indicating that the Cog-
nate Facilitation Effect was larger for English (178.3 ms) than Dundonian Scots (63.4 ms). This
effect corresponds with the mixed language block pattern observed in some bilingual studies
(e.g., [11,32]), which has been explained in terms of the more dominant language variant
being more susceptible to the influence of the less dominant language in mixed language
blocks. Along the lines of Experiment 1, we did not find a significant interaction between CSI
Length and Cognate Status (see Fig 2), indicating that the cognate facilitation effect was also
not substantially reduced by the longer preparation time for bidialectals.

An additional analysis with Trial type (switch vs. repetition trial) included showed worse
performance in switch (1057 ms) than repetition (1024 ms) trials (i.e., switch costs), $b = 18.6$,
$SE = 4.8$, $t = 3.9$, $p < .001$, but Trial type did not significantly interact with any other factors
(all $p$'s $> .15$).

## General discussion

In this study, we investigated whether proactive language control relies on inhibition during
bilingual and bidialectal language production. To this end, we manipulated the CSI and cog-
nate status. While cognate words were produced faster than non-cognate words by both bilin-
guals (Experiment 1) and bidialectals (Experiment 2), this effect did not interact with CSI
length in either experiment.

So, our results are similar to the behavioral results of Branzi et al. [25], who also did not
observe a significant interaction between CSI and cognate status with Spanish-Basque-English
multilinguals. If proactive language control relies on inhibition, one would expect to see less
cross-language activation with a long preparation time, since the long preparation time would
allow for more proactive inhibition of the non-target language. Consequently, a smaller cog-
nate facilitation effect would have been expected with a long CSI since the non-target language
would be inhibited. This was not the case in our study and in the behavioral results of Branzi
and colleagues. Together, these results could be interpreted as proactive language control not
relying on inhibition, both during bilingual and bidialectal language production.

It should be noted that Branzi et al. [25] did observe fMRI evidence for proactive inhibitory
control. More specifically, the bilateral anterior ventrolateral prefrontal cortex showed a reduc-
tion in the long versus short CSI for the cognate effect. So, one possibility is that behavioral
results are not sensitive enough to capture the target interaction of this study. Future research
based on different techniques (e.g., EEG, MEG, and pupil size) will have to indicate whether
this is the case.

Another possibility for why we did not observe a significant interaction between CSI and
cognate status is that our study did not have enough power to capture this interaction (cf.
[58]). However, when combining both datasets, and thus doubling the statistical power, the
interaction still did not reach significance, $b = 0.36$, $SE = 3.78$, $t = .09$, $p = .93$.

To provide statistical evidence in favor of the null hypothesis in both experiments, we addi-
tionally relied on Bayesian Null Hypothesis Testing analyses (e.g., [59,60]). This is a statistical
test that allows us to estimate the degree to which the Null hypothesis ($H_0$) should be accepted
over the Alternative hypothesis ($H_1$). Using this type of analysis, we compared a model that
included both main effects of CSI and Cognate Status and their interaction against a model
that included these two main effects without their interaction in JASP [61]. The results of the
Bayesian Null Hypothesis Testing on the data of Experiment 1 and Experiment 2 showed that
the model without the interaction accounts better for the data than the model with the

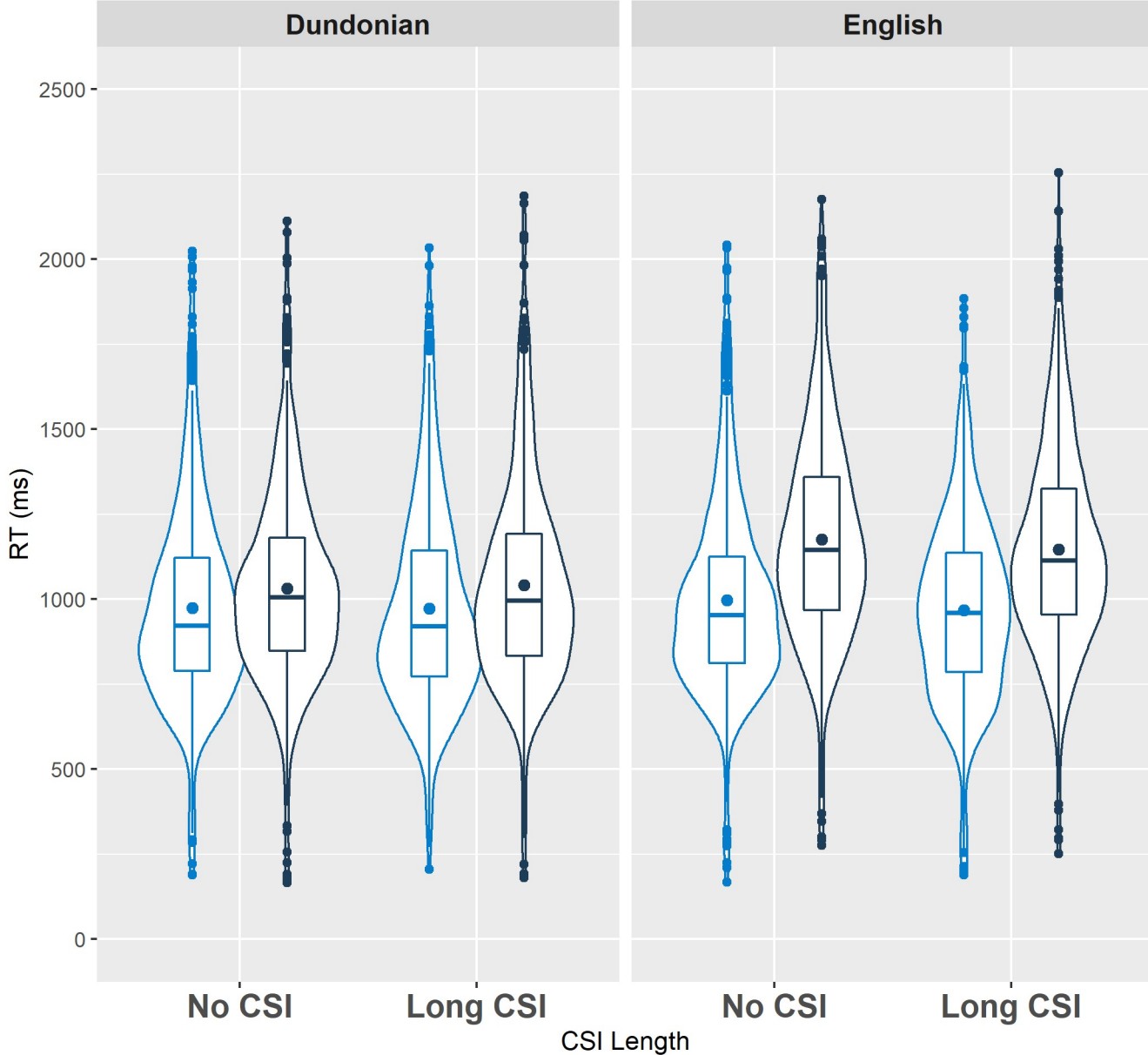

**Fig 2. Violin plot showing the distribution of cognate and non-cognate RTs in each CSI block across both varieties for Experiment 2 (Dundonian/English "bidialectals").** The boxplots show the interquartile range, the horizontal line represents the median, and the dot indicates the mean for each condition.

interaction (Experiment 1: $BF_{01} = 4.4$; Experiment 2: $BF_{01} = 4.2$; Joint Analysis: $BF_{01} = 5.9$). Put differently, based on the Bayesian Null Hypothesis Testing, we have statistical evidence that the cognate facilitation effect is about four times more likely to be similar across the long CSI and no CSI conditions than that the cognate facilitation effect is different across these two conditions, and almost six times more likely when combining both data sets (likely due to the

increase in statistical power). These results provide some evidence that we found a true null effect for the interaction between CSI Length and Cognate Status.

Taken together, the current study investigated the possibility of proactive inhibitory control during both bilingual and bidialectal language production through manipulations of the CSI and the cognate status of words. Along the results of some, but not all, previous studies that investigated this issue, no evidence for proactive inhibitory control was found with either bilinguals or bidialectals.

## Appendix

Word list of Experiment 1.

| Cognates | | Non-cognates | |
|---|---|---|---|
| Dutch | English | Dutch | English |
| appel | apple | aardbei | strawberry |
| arm | arm | auto | car |
| bal | ball | been | leg |
| banaan | banana | bloem | flower |
| bed | bed | boom | tree |
| boek | book | bril | glasses |
| boot | boat | broek | pants |
| brood | bread | eend | duck |
| bus | bus | fles | bottle |
| deur | door | geld | money |
| diamant | diamond | haai | shark |
| gitaar | guitar | halsketting | necklace |
| glas | glass | heks | witch |
| hand | hand | hond | dog |
| hart | heart | horloge | watch |
| helm | helmet | jurk | dress |
| huis | house | kaars | candle |
| kat | cat | kerk | church |
| lamp | lamp | kikker | frog |
| maan | moon | knop | button |
| muis | mouse | konijn | rabbit |
| neus | nose | krant | newspaper |
| peer | pear | mes | knife |
| piano | piano | oog | eye |
| pijp | pipe | paard | horse |
| piraat | pirate | paddestoel | mushroom |
| pizza | pizza | pop | doll |
| racket | rocket | raam | window |
| robot | robot | riem | belt |
| schoen | shoe | sleutel | key |
| sigaret | cigarette | spiegel | mirror |
| ster | star | stoel | chair |
| tijger | tiger | touw | rope |
| trein | train | varken | pig |
| wolf | wolf | vogel | bird |
| zon | sun | wortel | carrot |

Word list of Experiment 2.

| Cognates | | Non-cognates | |
|---|---|---|---|
| Dundonian Scots | English | Dundonian Scots | English |
| coo | cow | baffies | slippers |
| ezz | eyes | bairns | children |
| fermer | farmer | brae | hill |
| glesses | glasses | laddie | boy |
| hert | heart | lassie | girl |
| hoose | house | lugs | ears |
| moose | mouse | oxter | armpit |
| sassijis | sausages | plook | spot |
| screwdrehver | screwdriver | tattie | potato |

## Author Contributions

**Conceptualization:** Mathieu Declerck, Elisabeth Özbakar, Neil W. Kirk.

**Data curation:** Mathieu Declerck, Elisabeth Özbakar, Neil W. Kirk.

**Formal analysis:** Mathieu Declerck, Elisabeth Özbakar, Neil W. Kirk.

**Funding acquisition:** Neil W. Kirk.

**Investigation:** Mathieu Declerck, Elisabeth Özbakar, Neil W. Kirk.

**Methodology:** Mathieu Declerck, Neil W. Kirk.

**Project administration:** Mathieu Declerck, Neil W. Kirk.

**Resources:** Neil W. Kirk.

**Supervision:** Mathieu Declerck.

**Writing – original draft:** Mathieu Declerck, Elisabeth Özbakar, Neil W. Kirk.

**Writing – review & editing:** Mathieu Declerck, Elisabeth Özbakar, Neil W. Kirk.

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
