## [Decision Letter · Decision Letter 0]

29 Jun 2021

PONE-D-21-16575

Is there proactive inhibitory control during bilingual and bidialectal language production?

PLOS ONE

Dear Dr. Kirk,

Thank you for submitting your manuscript to PLOS ONE. After careful consideration, we feel that it has merit but does not fully meet PLOS ONE’s publication criteria as it currently stands. Therefore, we invite you to submit a revised version of the manuscript that addresses the points raised during the review process.

Two reviewers, both experts in the field of bilingualism, have now reviewed your manuscript. Both find this a relevant and well-designed study with interesting results, in particular with regard to bidialectal speakers (and I agree). Both reviewers do have several comments in particular with regard to the theoretical interpretations of the data. Both reviewers give clear and very constructive comments which will help to improve this interesting work. Based on the reviewers' feedback I have decided 'major revision', though I have little doubt the authors should be able to address the comments given by the reviewers.

We look forward to receiving your revised manuscript.

Kind regards,

Kristof Strijkers, Ph.D

Academic Editor

PLOS ONE

Additional Editor Comments:

Dear authors,

Two reviewers, both experts in the field of bilingualism, have now reviewed your manuscript. Both find this a relevant and well-designed with interesting results, in particular with regard to bidialectal speakers (and I agree). Both reviewers do have several comments in particular with regard to the theoretical interpretations of the data. Both reviewers give clear and very constructive comments which will help to improve this interesting work. Based on the reviewers' feedback I have decided 'major revision', though I have little doubt the authors should be able to address the comments given by the reviewers.

Sincerely,

Kristof

Journal Requirements:

2. Please change "female” or "male" to "woman” or "man" as appropriate, when used as a noun (see for instance https://apastyle.apa.org/style-grammar-guidelines/bias-free-language/gender).

Reviewers' comments:

Reviewer's Responses to Questions

**Comments to the Author**

1. Is the manuscript technically sound, and do the data support the conclusions?

Reviewer #1: Yes

Reviewer #2: Yes

2. Has the statistical analysis been performed appropriately and rigorously? 

Reviewer #1: Yes

Reviewer #2: Yes

3. Have the authors made all data underlying the findings in their manuscript fully available?

Reviewer #1: Yes

Reviewer #2: No

4. Is the manuscript presented in an intelligible fashion and written in standard English?

Reviewer #1: Yes

Reviewer #2: Yes

5. Review Comments to the Author

Reviewer #1: The manuscript reports two experiments that aim to investigate the role of proactive control in bilingual and bidialectal language production. The results of the experiments support the hypothesis that proactive language production is not dependent on inhibitory control. The results are especially relevant for the bidialectal language production.

I have some comments about some theoretical interpretations and some results, especially for Experiment 2.

1. Introduction. Language-blocked designs may differ from language-mixed ones in terms of proactive control (and inhibition). Possibly, proactive control in language-mixed designs arises from the combination of inhibition/activation of language (global level) and at lexical/local level (especially in switch trials). In language-blocked designs, as no language switching is required, inhibition at local/lexical level should be less involved (or at all). In the Introduction the effects that come from these two types of designs are discussed as they were the same (or depending on the same underlying mechanism), but I don’t think they are.

2. Introduction, CSI and cognate status. Longer preparation times have the effect of giving participants more time to reconfigure the new language in switch trials, but why should this be only for cognate words. If in this condition there is less activation of the non-target language, then naming latencies of both word categories (cognates and no-cognates) should decrease. So, the within-language facilitation (cognates vs. no-cognates) could be the same for short and long CSIs.

3. Introduction. It is assumed that proactive control is inhibition, however, evidence that supports this claim is very limited. Is there any alternative explanation to the inhibitory control hypothesis?

4. Comparison between bidialectal and bilingual language production. Is there any hypothesis that suggests that these two language systems may work differently in language switching? If the same linguistic or cognitive processes are universal, language pairs should not matter. Language pairs should be important only if linguistic variables (such as distance or typology) should have an impact on the mechanisms.

5. Language groups, Experiment 1 and 2. It seems that language usage is less balanced in Experiment 1 than in Experiment 2. Have you checked whether the percentages are different between the two experiments? The results seem to be not affected by this variable, but an imbalanced usage of the two languages, at least at theoretical level, may produce changes in the costs (asymmetrical vs. symmetrical).

6. Number of trials analyzed. 30% of RT data were excluded (26.5% as outliers and 3.6% as errors). Was the distribution of errors the same according to the trial types in both CSI conditions and for cognates and no-cognates? This is useful to check to exclude that one condition had more switch trials excluded or higher variability than the other.

7. Results of Experiment 2. Variety was statistically significant and significantly interacted with cognate status. Since this variable seems to have some relevance, it would be interesting to plot the data in both figures as a function of this factor as well.

8. Results, Experiment 1 and 2. Switch costs. From the data reported in the footnotes, the magnitude of the switch costs is 33 ms in Experiment 2 and 72 ms in Experiment 1. Is this reduction consistent with previous data from bidialectal language switching?

9. Discussion, CSI. The results from Experiment 2 showed that language preparation time is not affecting proactive control. How is this explained in terms of proactive control and language preparation?

10. Discussion, cognate status.

a. The magnitude of the cognate effect doubled in Experiment 2 (122 ms) as compared to Experiment 1 (54 ms) (beta values also suggest the same trend). Is there any explanation of this difference between the two experiments?

b. The cognate effect was larger in English (178 ms) than in Dundonian Scots (63 ms). Any explanation of this difference?

Reviewer #2: This manuscript presents two experiments addressing proactive control in bilinguals (exp 1) and bidialectal speakers (exp 2). The manuscript is very well written and easy to comprehend. The questions posed are of theoretical interest and novel (especially with respect to the bidialectal speakers). The experiments have been designed and analysed carefully. Most of my comments concern further discussion of theories/interpretations in the introduction and discussion:

- An important point throughout the manuscript seems to be that looking at cognate effects and CSI is a measure of proactive inhibitory control (i.e., proactive control by suppressing the non-target language) rather than proactive control by e.g., activating or prioritising the target language. This idea, however, is not really developed in the introduction. Why would an interaction between CSI and cognate facilitation support proactive inhibition specifically (and not proactive activation of the target language)? I am not questioning this interpretation but a more concrete explanation is needed.

- In both experiments a lot of data points had to be removed (26% and 22%). Error rates seem to be low, although no-responses do not seem to be counted as part of the error rate? What was the main reason so many trials had to be removed? Was it mainly participants not responding at all or was it related to the quality of the recordings?

- Related to the previous point, did you manually check how well Chronset determined naming onset times? I have found with previous work in the lab and online that Chronset's performance can really depend on the participant and the microphone used. It would be good to perhaps score a small number of trials for each participants manually and to report how well that aligned with Chronset.

- It's great to see that the OSF page includes a pre-registration, the materials, and information about how to score the accuracy. I wasn't able, however, to find the actual data. I'm not sure if I just couldn't find the folder or if the data file is set to private?

- Footnotes explain some further analyses with trial type, but trial type was not included in the main analyses/pre-registration. It might be worth explaining why trial type was not included.

- In the discussion it would be good to see some evaluation of the main effect of CSI, which was found in Exp 1 but not in Exp 2. Was the chosen CSI sufficient to show any effects of "preparation" time? It would help to explain how this CSI was chosen and how other studies manipulating CSI have shown that it can influence e.g., task or language switching and mixing effects.

- In a task in which half of the items are cognates, would proactively inhibiting one of the languages be the most "fruitful" approach? I am just leaving this here as a point to consider (and perhaps cover in the discussion), but if half of your responses are actually facilitated by having both languages active (and you don't know if that'll be the case until you see the actual picture), you might not want to inhibit the non-target language, even when you have more time to respond to the cue.

- More generally, it might help to present the different measures of proactive control that have been used in the literature and to refer back to those different measures in the discussion. This would help to place this study more clearly in the literature on this topic. In the discussion it would also be good to see a more in-depth evaluation of previous studies that have or have not found evidence for proactive control (whether inhibitory or not) in bilingual and bidialectal speakers. For example, the introduction to Experiment 2 mentions a 2021 study reporting mixing costs as a measure of proactive control in bidialectal speakers. Linking the current study to those previous findings in the discussion section would strengthen the interpretation of the study.

Minor point:

p 4 states that: "The behavioral pattern in this setup entails worse performance in Block 1 than in Block 3". Is this correct? There wasn't a reference for this description but would you not expect worse performance in Block 3?

6. PLOS authors have the option to publish the peer review history of their article (what does this mean?). If published, this will include your full peer review and any attached files.

Reviewer #1: **Yes: **Marco Calabria

Reviewer #2: No

---

## [Author Response · Author response to Decision Letter 0]

3 Aug 2021

Dear Dr. Strijkers,

I am sending you the revised version of our manuscript named “Is there proactive inhibitory control during bilingual and bidialectal language production?”. We are grateful for the comments by the reviewers and incorporated them where possible. 

A detailed point-to-point response to the questions of the reviewers is provided below. Where appropriate a reference to the respective changes in the manuscript is given and the major changes are highlighted in the manuscript (word count of manuscript: 5,013).

Reviewer 1

1. Introduction. Language-blocked designs may differ from language-mixed ones in terms of proactive control (and inhibition). Possibly, proactive control in language-mixed designs arises from the combination of inhibition/activation of language (global level) and at lexical/local level (especially in switch trials). In language-blocked designs, as no language switching is required, inhibition at local/lexical level should be less involved (or at all). In the Introduction the effects that come from these two types of designs are discussed as they were the same (or depending on the same underlying mechanism), but I don’t think they are.

Reply: It is unclear why proactive language control should occur at the local/lexical level in mixed language blocks but not in single language blocks. Moreover, it is also not clear why this restriction does not apply to proactive language control on the language level. The reviewer indicates that the lack of language switches in single language blocks is the reason for no proactive language control on the lexical level in single language blocks. However, even in the blocked language order paradigm there are language switches from one single language block to another. Furthermore, proactive language control is implemented prior to any cross-language interference as an anticipatory control processes for said cross-language interference. So, if anything, proactive language control might be higher because more language switches might be involved in mixed language blocks, but no qualitative differences are necessarily involved.

 It is not even clear whether proactive language control occurs at the lexical/local level at all. For proactive language control to occur at the lexical level, one would have to assume that processing a specific item will lead to an alteration of proactive language control for that specific item while processing it, but not for other items. This sounds more like reactive language control.

 However, to accommodate the reviewer’s comment, we made it clear that in the blocked language order paradigm the proactive language control is implemented in single language blocks (page 4), whereas in the study of Branzi et al. (2020), proactive language control in mixed language blocks was investigated (pages 5-6). This should make any possible distinctions between proactive language control in these different linguistic settings clear. 

2. Introduction, CSI and cognate status. Longer preparation times have the effect of giving participants more time to reconfigure the new language in switch trials, but why should this be only for cognate words. If in this condition there is less activation of the non-target language, then naming latencies of both word categories (cognates and no-cognates) should decrease. So, the within-language facilitation (cognates vs. no-cognates) could be the same for short and long CSIs. 

Reply: We did not explain the rationale behind our, and Branzi et al.’s (2020), manipulation well in the original submission. We do not assume that the proactive language control, afforded by a long CSI, will only affect cognate words. However, because cognates are produced faster due to the non-target language being activated, proactive inhibition of the non-target language implemented in the long CSI condition should reduce that non-target language activation. Hence, if proactive language control relies on inhibition, then we expected a smaller cognate facilitation effect in the long CSI condition than in the condition when there was no CSI. We have made this clearer in the introduction (page 6).

3. Introduction. It is assumed that proactive control is inhibition, however, evidence that supports this claim is very limited. Is there any alternative explanation to the inhibitory control hypothesis?

Reply: We have now included an alternative activation account for proactive language control in the context of the reversed language dominance effect and the blocked language order effect (pages 4 and 5-6).

4. Comparison between bidialectal and bilingual language production. Is there any hypothesis that suggests that these two language systems may work differently in language switching? If the same linguistic or cognitive processes are universal, language pairs should not matter. Language pairs should be important only if linguistic variables (such as distance or typology) should have an impact on the mechanisms.

Reply: We have several reasons to not automatically assume that language control is identical across bilinguals and bidialectals. For instance, language control does not operate in a similar way across modalities (e.g., Blanco-Elorrieta & Pylkkänen, 2016) or even language pairs (e.g., Kaufman et al., 2018; Prior & Gollan, 2011). Interesting for the current study is that a recent bilingual study even showed that more similar language pairs can result in an absence of language control (Deibel, 2020). We have included this information in the revised manuscript (page 15).

5. Language groups, Experiment 1 and 2. It seems that language usage is less balanced in Experiment 1 than in Experiment 2. Have you checked whether the percentages are different between the two experiments? The results seem to be not affected by this variable, but an imbalanced usage of the two languages, at least at theoretical level, may produce changes in the costs (asymmetrical vs. symmetrical).

Reply: We compared the % use of Dutch (Experiment 1) and Dundonian (Experiment 2) as a proxy of balancing and found that Dutch was used more often (65%) than Dundonian (42%), t(75) = 3.206, p = .002. We added this as an additional difference between the two participant groups, and thus a way to generalize the findings of Experiment 1 in Experiment 2 (page 15).

6. Number of trials analyzed. 30% of RT data were excluded (26.5% as outliers and 3.6% as errors). Was the distribution of errors the same according to the trial types in both CSI conditions and for cognates and no-cognates? This is useful to check to exclude that one condition had more switch trials excluded or higher variability than the other.

Reply: A total of 26.5% of trials in Experiment 1 (not 30%) and 22.7% in Experiment 2 were excluded. This included both outliers and errors.

Below you can find the distribution of the error rate per CSI and Cognate status condition (i.e., the conditions of interest) for Experiments 1 and 2 (Table R1). The distribution of the excluded trials per CSI and Cognate status for Experiments 1 and 2 are also provided below (Table R2). We did not include Trial Type because this factor was not in our main analysis, as was outlined in the pre-registration. We only mentioned this factor as an exploratory analysis to ensure it would not affect our main interaction of interest (CSI Block x Cognate Status; see also Comment 4 of Reviewer 2).

Table R1. Overall mean error rate in percentages (SD in parenthesis) as a function of CSI Length (No vs. Long CSI) and Cognate Status (cognates vs. non-cognates) for Experiments 1 and 2.

 Experiment 1 Experiment 2

Trial type Cognates Non-cognates Cognates Non-cognates

No CSI 2.6 (3.9) 4.9 (4.6) 3.1 (3.3) 3.6 (5.3)

Long CSI 3.6 (5.8) 3.8 (4.4) 3.3 (3.9) 5.1 (5.6)

Table R2. Overall mean excluded trials in percentages (SD in parenthesis) as a function of CSI Length (No vs. Long CSI) and Cognate Status (cognates vs. non-cognates) for Experiments 1 and 2.

 Experiment 1 Experiment 2

Trial type Cognates Non-cognates Cognates Non-cognates

No CSI 23.2 (10.9) 31.5 (13.9) 22.0 (13.5) 20.6 (15.4)

Long CSI 23.1 (11.8) 28.2 (13.1) 26.1 (14.7) 22.2 (15.0)

We believe that any differences (see the tables above), which are to be expected as the CSI length and cognate status could also influence the error rates and thus also the overall excluded trials, between conditions did not substantially influence our results, as we had a relatively large number of trials to start with. Additionally, when combining the data of both experiments, we found the same pattern (see page 23). This also shows that regardless of the number of trials per condition, the same pattern emerges.

7. Results of Experiment 2. Variety was statistically significant and significantly interacted with cognate status. Since this variable seems to have some relevance, it would be interesting to plot the data in both figures as a function of this factor as well.

Reply: We have recreated both figures to incorporate this suggestion and have added them to the manuscript (pages 14 and 21). 

8. Results, Experiment 1 and 2. Switch costs. From the data reported in the footnotes, the magnitude of the switch costs is 33 ms in Experiment 2 and 72 ms in Experiment 1. Is this reduction consistent with previous data from bidialectal language switching?

Reply: As far as we can tell, the size of our bidialectal switch costs is in the same ballpark as the switch costs observed in Vorweg et al. (2019) and Scaltritti et al. (2017). However, it is much lower than the 111 ms switch costs observed in Kirk et al. (2021). We assume that this is due to the Dundonian dialect used in the current study being less geographically widespread and occupying a lower social status, thus Dundonian bidialectals are accustomed to switching between their two language varieties regularly since they can’t use their dialect with everybody and in all aspects of day-to-day life. The Orcadian bidialectals from Kirk et al. (2021), on the other hand, do not have to switch often between their language varieties, because the Orcadian dialect is spoken ubiquitously throughout the Orkney islands, and is given more social prestige. From bilingual studies we know that regular language switching can reduce language switch costs (e.g., Prior & Gollan, 2011).

9. Discussion, CSI. The results from Experiment 2 showed that language preparation time is not affecting proactive control. How is this explained in terms of proactive control and language preparation?

Reply: We added an interpretation of this absence of a CSI effect in the revised manuscript (pages 18-19).

10. Discussion, cognate status.

a. The magnitude of the cognate effect doubled in Experiment 2 (122 ms) as compared to Experiment 1 (54 ms) (beta values also suggest the same trend). Is there any explanation of this difference between the two experiments?

b. The cognate effect was larger in English (178 ms) than in Dundonian Scots (63 ms). Any explanation of this difference?

Reply: a. It might be that there is a larger phonological overlap between the cognates in Experiment 2, as more phonemes are shared between Dundonian Scots and Standard English than between Dutch and English. This could explain the larger cognate facilitation effect with bidialectals than with bilinguals. However, because we did not match the items across both experiments, it could be that any number of item characteristics that differed between the two experiments led to the larger cognate facilitation effect in Experiment 2. 

b. The larger cognate facilitation effect for the more dominant language variant in mixed language blocks has also been observed in some bilingual studies (e.g., Christoffels et al., 2007; Verhoef et al., 2009). We have included this, and the explanation for this effect in bilingual studies, to the revised manuscript (page 19).

Reviewer 2

1. An important point throughout the manuscript seems to be that looking at cognate effects and CSI is a measure of proactive inhibitory control (i.e., proactive control by suppressing the non-target language) rather than proactive control by e.g., activating or prioritising the target language. This idea, however, is not really developed in the introduction. Why would an interaction between CSI and cognate facilitation support proactive inhibition specifically (and not proactive activation of the target language)? I am not questioning this interpretation but a more concrete explanation is needed.

Reply: We have explained our manipulation and its relation to proactive inhibitory control more in the revised manuscript (page 6; see also Comment 2 of Reviewer 1).

2. In both experiments a lot of data points had to be removed (26% and 22%). Error rates seem to be low, although no-responses do not seem to be counted as part of the error rate? What was the main reason so many trials had to be removed? Was it mainly participants not responding at all or was it related to the quality of the recordings?

did you manually check how well Chronset determined naming onset times? I have found with previous work in the lab and online that Chronset's performance can really depend on the participant and the microphone used. It would be good to perhaps score a small number of trials for each participants manually and to report how well that aligned with Chronset.

Reply: Using an online platform without an experimenter present means it is less clear if a “no-response” is down to the participant not producing a response in the timeframe or whether it’s due to a recording error. For that reason, we did not mark “no-response” trials as incorrect unless it was clear that the recording had actually captured the participant not giving a response (i.e. we could hear breathing or muttering, indicating that it was not a recording failure). We therefore filtered out trials that we had marked as “no sound” before we conducted the error analyses, which probably accounts for the low number of errors, yet a high number of exclusions. There may also be occasions when Chronset did not detect a response even though we could hear a correct response being given, which may account for some additional exclusions based on our criteria of excluding RTs under 150ms. We did not initially manually check naming onset time. However, based on this useful suggestion we have now checked a random sample of 5% of trials from each participant in both experiments and measured these manually using Praat (over 400 trials in total). We ran a Pearson correlation between the Chronset and manual RTs which showed a strong correlation (r(419) = .921, p <.001), which is in line with the correlations reported by Roux et al. (2017) in their paper introducing the Chronset platform. This information has been added to the revised manuscript (page 11).

3. It's great to see that the OSF page includes a pre-registration, the materials, and information about how to score the accuracy. I wasn't able, however, to find the actual data. I'm not sure if I just couldn't find the folder or if the data file is set to private?

Reply: Sorry about this! We thought that putting the main page as public would make the entire project public, but that was not the case. We now also put the separate links for each experiment as public. Thank you for pointing this out! 

4. Footnotes explain some further analyses with trial type, but trial type was not included in the main analyses/pre-registration. It might be worth explaining why trial type was not included.

Reply: We did include in the pre-registration that we might include trial type (see section “Other” in the pre-registration). As we suggest in the pre-registration, we did not include this factor into the main analysis because we were not really interested in how our main interaction of interest (i.e., CSI Length x Cognate Status) was influenced by trial type. We just included this additional analysis with the factor trial type to make sure that this factor did not hide any significant effects of said interaction of interest. We have further elaborated on this in the revised manuscript (page 12). 

5. In the discussion it would be good to see some evaluation of the main effect of CSI, which was found in Exp 1 but not in Exp 2. Was the chosen CSI sufficient to show any effects of "preparation" time? It would help to explain how this CSI was chosen and how other studies manipulating CSI have shown that it can influence e.g., task or language switching and mixing effects.

Reply: We chose a long CSI of 1250ms, identical to Fink and Goldrick (2015) and Verhoef et al. (2009). This long CSI length is on the longer side relative to previous studies, which relied on a long CSI of 800ms (e.g., Costa & Santesteban, 2004; Ma et al., 2016) or around 1000ms (e.g., Branzi et al., 2020; Philipp et al., 2007; Stasenko et al., 2017). Yet, it is not the longest, as Lavric and colleagues used a 1500ms CSI. Hence, this CSI should have been long enough to influence performance. Regarding the short CSI, we actually chose to have no CSI (i.e., 0ms CSI), like Declerck, Ivanova et al. (2020) and Mosca and Clahsen (2016). We chose this approach as it would provide a more extreme difference between preparation and no preparation than more and less preparation. The combination of the relatively long CSI compared to the 0 ms CSI should have been sufficient to see any preparation time effect, as was the case in Experiment 1. This information has been added to the revised manuscript (page 8).

 While we already discussed the main effect of CSI in Experiment 1 of the original manuscript (page 12), in the revised manuscript we now also discuss the lack of a main effect of CSI in Experiment 2 (pages 18-19; see also Comment 9 of Reviewer 1). 

6. In a task in which half of the items are cognates, would proactively inhibiting one of the languages be the most "fruitful" approach? I am just leaving this here as a point to consider (and perhaps cover in the discussion), but if half of your responses are actually facilitated by having both languages active (and you don't know if that'll be the case until you see the actual picture), you might not want to inhibit the non-target language, even when you have more time to respond to the cue.

Reply: This is a good point that we had not considered! However, we do not think that the cognitive system can decide to forgo the processes that are supposed to typically optimize performance (like language control) to make a specific condition as optimal as possible. Even if the cognitive system was able to decide the most optimal way like this, would it choose to rely on a suboptimal processing procedure (i.e., no proactive language control) for the more difficult items (i.e., noncognates)?

7. More generally, it might help to present the different measures of proactive control that have been used in the literature and to refer back to those different measures in the discussion. This would help to place this study more clearly in the literature on this topic. In the discussion it would also be good to see a more in-depth evaluation of previous studies that have or have not found evidence for proactive control (whether inhibitory or not) in bilingual and bidialectal speakers. For example, the introduction to Experiment 2 mentions a 2021 study reporting mixing costs as a measure of proactive control in bidialectal speakers. Linking the current study to those previous findings in the discussion section would strengthen the interpretation of the study.

Reply: The introduction now starts off with the presentation of two measures of proactive language control and their inhibition and activation accounts: the reversed language dominance effect and the blocked language order effect (pages 3-5). We did not include mixing costs here, as this measure can also be explained with additional/alternative processes. The latter information has also been added to the revised manuscript (pages 16 and 19).

 We did not include an in-depth evaluation on whether previous studies have observed proactive language control through the several effects that have been linked to this process, as this would be beyond the scope of the current study. We did cite a review article on proactive language control that discusses previous studies that did or did not find effects related to proactive language control (cf. Declerck, 2020; see pages 3 and 16).

Since, as far as we know, no previous studies have evaluated proactive language control in bidialectals, we did decide to discuss the relation between our current findings and the mixing costs with bidialectals found in a previous study (Kirk et al., 2021). This discussion can be found in the Results and Discussion section of Experiment 2 (pages 18-19), as we wanted to keep the main focus of the General Discussion on whether proactive language control relies on inhibition.

8. p 4 states that: "The behavioral pattern in this setup entails worse performance in Block 1 than in Block 3". Is this correct? There wasn't a reference for this description but would you not expect worse performance in Block 3?

Reply: Thank you for pointing this error out! We have corrected this and added references for this blocked language order setup and the other blocked language order setup (page 4).

We would like to thank you and the reviewers for the helpful comments. We hope you find our revision and response to the reviewers’ comments satisfactory.

Sincerely,

Mathieu Declerck, Elisabeth Özbakar, and Neil W. Kirk

---

## [Decision Letter · Decision Letter 1]

31 Aug 2021

Is there proactive inhibitory control during bilingual and bidialectal language production?

PONE-D-21-16575R1

Dear Dr. Kirk,

We’re pleased to inform you that your manuscript has been judged scientifically suitable for publication and will be formally accepted for publication once it meets all outstanding technical requirements.

Kind regards,

Kristof Strijkers, Ph.D

Academic Editor

PLOS ONE

Additional Editor Comments (optional):

Dear authors,

I have send your revised manuscript to the same two reviewers of the original submission and I am glad to say that both recommend that your manuscript be accepted, and I am happy to follow that recommendation.

Congratulations,

Kristof Strijkers

Reviewers' comments:

Reviewer's Responses to Questions

**Comments to the Author**

1. If the authors have adequately addressed your comments raised in a previous round of review and you feel that this manuscript is now acceptable for publication, you may indicate that here to bypass the “Comments to the Author” section, enter your conflict of interest statement in the “Confidential to Editor” section, and submit your "Accept" recommendation.

Reviewer #1: All comments have been addressed

Reviewer #2: All comments have been addressed

2. Is the manuscript technically sound, and do the data support the conclusions?

Reviewer #1: Yes

Reviewer #2: Yes

3. Has the statistical analysis been performed appropriately and rigorously? 

Reviewer #1: Yes

Reviewer #2: Yes

4. Have the authors made all data underlying the findings in their manuscript fully available?

Reviewer #1: Yes

Reviewer #2: Yes

5. Is the manuscript presented in an intelligible fashion and written in standard English?

Reviewer #1: Yes

Reviewer #2: Yes

6. Review Comments to the Author

Reviewer #1: I would like to thank the authors for their responses to my comments. I consider that the authors have adequately responded to the comments raised in the first review, and hence I consider this manuscript suitable for publication.

Reviewer #2: I would like to thank the authors for their revisions. My previous comments have all been addressed.

7. PLOS authors have the option to publish the peer review history of their article (what does this mean?). If published, this will include your full peer review and any attached files.

Reviewer #1: **Yes: **Marco Calabria

Reviewer #2: No

---

## [Editor Report · Acceptance letter]

2 Sep 2021

PONE-D-21-16575R1 

Is there proactive inhibitory control during bilingual and bidialectal language production? 

Dear Dr. Kirk:

I'm pleased to inform you that your manuscript has been deemed suitable for publication in PLOS ONE. Congratulations! Your manuscript is now with our production department. 

Kind regards, 

on behalf of

Dr. Kristof Strijkers 

Academic Editor

PLOS ONE